# Retinoids Delivery Systems in Cancer: Liposomal Fenretinide for Neuroectodermal-Derived Tumors

**DOI:** 10.3390/ph14090854

**Published:** 2021-08-26

**Authors:** Veronica Bensa, Enzo Calarco, Elena Giusto, Patrizia Perri, Maria Valeria Corrias, Mirco Ponzoni, Chiara Brignole, Fabio Pastorino

**Affiliations:** Laboratory of Experimental Therapies in Oncology, IRCCS Istituto Giannina Gaslini, 16147 Genoa, Italy; veronicabensa@gaslini.org (V.B.); enzocalarco@gaslini.org (E.C.); ele0696@alice.it (E.G.); perripatrizia@gaslini.org (P.P.); mariavaleriacorrias@gaslini.org (M.V.C.); mircoponzoni@gaslini.org (M.P.)

**Keywords:** retinoids, fenretinide (4-HPR), nanotechnology, liposomes

## Abstract

Retinoids are a class of natural and synthetic compounds derived from vitamin A. They are involved in several biological processes like embryogenesis, reproduction, vision, growth, inflammation, differentiation, proliferation, and apoptosis. In light of their important functions, retinoids have been widely investigated for their therapeutic applications. Thus far, their use for the treatment of several types of cancer and skin disorders has been reported. However, these therapeutic agents present several limitations for their widespread clinical translatability, i.e., poor solubility and chemical instability in water, sensitivity to light, heat, and oxygen, and low bioavailability. These characteristics result in internalization into target cells and tissues only at low concentration and, consequently, at an unsatisfactory therapeutic dose. Furthermore, the administration of retinoids causes severe side-effects. Thus, in order to improve their pharmacological properties and circulating half-life, while minimizing their off-target uptake, various retinoids delivery systems have been recently developed. This review intends to provide examples of retinoids-loaded nano-delivery systems for cancer treatment. In particular, the use and the therapeutic results obtained by using fenretinide-loaded liposomes against neuroectodermal-derived tumors, such as melanoma, in adults, and neuroblastoma, the most common extra-cranial solid tumor of childhood, will be discussed.

## 1. Introduction

Retinoids are a family of compounds related to vitamin A (retinol), essential for the life of all chordates. They are signaling molecules that, after binding to the nuclear retinoic acid receptors (RARs) and retinoic X receptors (RXRs), activate genetic networks involved in important biological and physiological processes, such as cell proliferation, cell differentiation, apoptosis, and fetal development [1,2,3]. The term “*retinoids”* was introduced in 1976 by Sporn and colleagues [4], and in 1981, the IUPAC-IUB Joint Commission on Biochemical Nomenclature (JCBN) defined retinoids: (i) compounds composed of four isoprenoid units joined head-to-tail; (ii) derived from a monocyclic molecule; (iii) containing five carbon-carbon double bonds and a functional terminal group at the end of the acyclic portion. According to this definition, the retinoids family includes the natural forms of vitamin A and their synthetic derivatives ([1] and http://publications.iupac.org) (accessed on 15 March 2021). To date, more than 1500 different related compounds have been discovered and tested. Retinoids have raised interest within the scientific community thanks to their beneficial effects in vision [5], skin disorders (acne, psoriasis, and keratinization disorders) [6], and cancer [7,8,9]. Specifically, in the oncology field, retinoids attracted researchers’ attention due to their known anti-tumor properties. In particular, they were demonstrated to be effective in inducing differentiation and/or apoptosis of tumor cells, as well as cell growth inhibition [10,11,12,13]. Moreover, they also showed chemo-preventive effects in experimental animal models of chemically-induced cancer [14]. The most frequently tested vitamin A derivatives in cancer medicine are represented by all-trans retinoic acid (ATRA, tretinoin), 9-cis retinoic acid (9-cis-RA, alitretinoin), and 13-cis retinoic acid (13-cis-RA, isotretinoin) [7]. Clinically, the most encouraging therapeutic results were obtained by the use of ATRA and 13-cis-RA following bone marrow transplantation in patients affected by acute promyelocytic leukemia and high-risk neuroblastoma (NB), respectively [15,16,17]. In contrast, the initial enthusiasm derived from several pre-clinical studies faded after their clinical application because of toxicity-driven limitations. Indeed, long-term and therapeutic-dosage administrations of natural retinoids caused liver toxicity, dry skin and irritation, bone damage, lipid alterations, and teratogenicity [7,18,19,20]. Further, retinoids are sensitive to oxygen, heat, and light and present poor solubility in water, all characteristics that reduce their bioavailability drastically and, consequently, their therapeutic efficacy [18]. Finally, they are characterized by a short lifetime due to the degradation by the cytochrome P450-dependent monooxygenase system [18]. Consequently, the development of less toxic and more bioavailable vitamin A-related compounds became necessary.

At present, there are four generations of retinoids differing from each other by the modifications of the vitamin A molecular structure [21,22,23] (Table 1). Among others, the synthetic derivative of ATRA, the amide analog of retinoic acid, N-(4-hydroxyphenyl)retinamide (fenretinide/4-HPR), belonging to the third generation of retinoids, deserved great attention (Table 1). Developed at the beginning to treat skin disorders, it was then investigated for its potential as a novel anti-cancer therapeutic. 4-HPR represented, indeed, one of the most promising drugs because of its favorable toxicological profile, characterized by minimal systemic toxicity, good tolerability, and high anti-tumor efficacy [24,25]. Due to its preferential accumulation in the mammary gland, it initially seemed very efficacious against breast cancer [26]. Successively, in vitro and pre-clinical experiments first and clinical trials later demonstrated that fenretinide was active against several types of cancers, including bladder, lung, ovary, prostate, melanoma, and NB [10,11,27]. Evidence shows that 4-HPR exerts anti-tumor effects on both premalignant cells by inhibiting the carcinogenesis process, and on transformed cells, by activating apoptosis, making fenretinide a promising compound for clinical application, both as a chemo-preventive agent and an anti-cancer drug [24,28].

A different strategy to reduce side-effects related to free retinoids administration while increasing their bioavailability and maximizing their therapeutic index is also represented by the design and development of appropriate Drug Delivery Systems. With this aim, several formulations loaded with retinoids have been developed [18,29].

## 2. Drug Delivery Systems for Cancer Therapy

Drug Delivery Systems (DDSs) refer to formulations able to transport and deliver active molecules/drugs to cell/tissue targets in order to achieve a specific and hopefully increased therapeutic effect compared to the free drug while minimizing its potential side effects. The use of DDSs in cancer therapy lies in the possibility of increasing the therapeutic index of the encapsulated drugs by delivering them to tumor cells through both passive and active targeting. Passive targeting of tumor cells exploits the so-called “Enhanced Permeability and Retention” effect (EPR) [30]. It is well known that the newly formed blood vessels of solid tumors present altered permeability, rendering them more permeable compared to those of healthy tissues [30]. In these circumstances, the leaky blood vessels allow for the non-selective extravasation of macromolecules (larger than 40 kDa) and small particles (ranging from 50 to 500 nm) into the tumor stroma, finally leading to tumor cells killing [30,31]. However, the passive targeting capability of DDSs only leads to a modest increased delivery of the encapsulated drug to the target site, and it is strictly dependent on different factors such as size and circulation time of the carrier, as well as on tumor biology features, such as vascularity of the tumor and leakiness of the vessels [31]. On the other hand, the passive targeting capability of DDSs can be further optimized by coupling moieties (e.g., monoclonal antibodies, peptides etc.) on their external surface, with the aim to specifically recognize and target tumor-associated antigens [18,31,32,33,34].

The most relevant DDSs used in pre-clinical studies of cancer therapy are nanodisks (NDs), polymeric micelles, dendrimers, and liposomes (Figure 1). In these formulations, retinoic acids can be either entrapped into the inner core or mixed with the outer surface [18]. Below, some examples of each mentioned retinoids carrying DDSs are presented.

### 2.1. Nanodisks (NDs)

NDs are self-assembled nanoscale carriers composed of a phospholipid bilayer surrounded by amphipathic apolipoproteins that stabilize the structure, serving as a scaffold [35,36] (Figure 1A). This composition allows the encapsulation and the delivery of hydrophobic molecules, such as amphotericin B and ATRA [35,37]. NDs have the advantages of being very small in size (8–20 nm in diameter) and fully soluble in water [36]. Singh and colleagues used NDs to encapsulate ATRA for treating cell culture models of mantle cell lymphoma (MCL). Compared to free ATRA, they demonstrated that ATRA-NDs were more effective in inducing MCL cells apoptosis and G1 cell cycle arrest in vitro [38]. Then, they optimized the formulation by adding the single chain variable antibody fragment against CD20 on the surface in order to improve the selective targeting of CD20-positive MCL cells. In this case, NDs were loaded with either ATRA (ATRA-NDs) or curcumin (curcumin-NDs) and the combination therapy was able to induce higher tumor apoptosis compared to each single treatment [38,39]. Importantly, these anti-CD20 NDs, although developed for treating MCL cells, may be useful for any other CD20-expressing tumors [39]. In another study, Buehler et al. engineered vault nanoparticles in order to encapsulate ATRA, using a vault-binding lipoprotein complex that creates a lipid bilayer NDs [40]. Testing hepatocellular carcinoma cell viability after ATRA-vaults treatment, they demonstrated that ATRA-NDs caused increased tumor cells killing compared to that obtained by free ATRA [40].

### 2.2. Polymeric Micelles

Polymeric Micelles are composed of amphiphilic polymers, which self-associate when added to an aqueous solvent. After self-assembly in the aqueous environment, the hydrophilic polymers (e.g., poly(ethylene glycol), chitosan, dextran, and hyaluronic acids) face the aqueous medium forming a hydrophilic shell, while the hydrophobic ones (e.g., poly(lactide) (PLA), poly(caprolactone) (PCL), poly(lactide-co-glycolide) (PLGA), polyesters, poly(amino acids), and lipids) form the hydrophobic core (Figure 1B). Similar to NDs, Polymeric Micelles can be employed to encapsulate hydrophobic drugs [41]. The anti-cancer agents can be conjugated to the distal ends of polymer to prepare pharmacologically active polymeric systems that enhance solubility and stability of the conjugates, providing an opportunity for combined drug delivery [41]. Specifically, an efficient intracellular drug delivery system is represented by the use of biocompatible polymeric micelles (BPMs), which allow the administration of retinoic acid (RA), protecting RA from metabolic deactivation while reducing RA-mediated toxicity [42]. For instance, the apoptotic effects induced by RA, either free or encapsulated into BPMs, were compared on colon cancer cell lines. When loaded into BPMs, RA led to a stronger effect with respect to the free administration, also despite the lower dose used [42]. Furthermore, Orienti et al. developed a nano-micellar formulation entrapping 4-HPR into the inner core, called bionanofenretinide (Bio-nFeR) [43]. This system increased fenretinide bioavailability, showing anti-tumor activity against lung, colon, and melanoma cancer stem cells, both in vitro and in tumor xenografts. Interestingly, Bio-nFeR showed lower toxicity when compared to NCI-FeR, an oral formulation of 4-HPR, consisting of soft gelatin capsules, actually available at the National Cancer Institute, and administered in clinical trials [43].

### 2.3. Dendrimers

Dendrimers are polymeric molecules composed of multiple repetitive branches arising radially from a central core. The terminal groups of every branch provide modifiable functionalities. The number of repeated branching units determines the generation of the dendrimer [44,45] (Figure 1C). Dendrimers are widely used as carriers for the delivery of several therapeutics compounds, including retinoids [46]. They present several advantageous features such as high water solubility, monodispersity, biocompatibility, and low immunogenicity [47]. Moreover, pH-sensitive formulations have been developed in order to be stable at physiological pH and to dissociate in the acid environment of the endosomal and lysosomal tumor compartments, resulting in an enhanced cellular uptake into target cells. For instance, Wang et al. synthesized pH-sensitive nanoparticles based on poly(amidoamine) (PAMAM) dendrimers encapsulating ATRA. They tested the formulation in vitro on human hepatocellular liver carcinoma cells, demonstrating its ability to arrest tumor cell proliferation and increase tumor cell death, compared to free ATRA [48]. Yalçın et al. loaded gemcitabine together with ATRA into PAMAM dendrimer-coated magnetic nanoparticles (DcMNPs) in order to simultaneously target gemcitabine-resistant pancreatic cancer cells and pancreatic stellate cells (PSC), stromal cells that support tumorigenesis, and form a fibrotic barrier against therapeutic agents [49,50]. They firstly proved that the DcMNPs were successfully internalized by pancreatic cancer cell lines and by primary human PSC. Then, overcoming pancreatic cancer cell’s resistance to gemcitabine, showed that the increased gemcitabine- and ATRA-loaded DcMNPs accumulation into tumor cells and tumor stroma caused a significant cell death compared to that obtained by ATRA or gemcitabine administered separately [49].

### 2.4. Liposomes

Liposomes are spherical-shaped vesicles composed of a hydrophilic aqueous space surrounded by one or more phospholipid bilayers, making them similar to the cell membrane structure [51,52] (Figure 1D). They can entrap both hydrophobic and hydrophilic compounds. The ability of liposomes to encapsulate “drugs” characterized by different solubility in water and to specifically target organs, tissues, and cells makes them attractive candidates for drug delivery [53]. They can be classified on the basis of: (i) size; (ii) lipid composition; (iii) surface modification. Due to their good features, such as biocompatibility, biodegradability, and low toxicity, liposomes are the first DDSs that have been translated to clinical application [54,55,56]. Further, they are the most frequently used formulations for drugs encapsulation and, at present, several liposomal formulations have been approved by the FDA, and different products are available for clinical application (e.g., Doxil^®^, Ambisome^®^, DepoDur™, DaunoXome^®^, etc. [54,55,56]). Moreover, liposomes are the only nanosystems used in clinical trials for the delivery of retinoids in solid cancer (https://clinicaltrials.gov/) (accessed on 15 March 2021). To date, pre-clinical evaluations of retinoids-encapsulating liposomes have been testing against several types of cancer, including lung, thyroid, and liver cancers, as well as on neuroectodermal-derived tumors such as melanoma and NB [57,58,59,60,61,62]. In particular, the anti-tumor effects of cationic liposomes encapsulating ATRA were also tested in pre-clinical animal models of lung cancer [57]. Interestingly, in this study aimed at investigating ATRA-driven reactivation of the tumor suppressor protein retinoic acid receptor beta (RAR-β), it was shown that, compared to free ATRA, the treatment with ATRA-loaded liposomes led to an enhanced RAR-β expression, thus becoming a useful molecular target therapy for lung cancer [57]. In another study, with the aim to reduce ATRA photo-degradation during administration as a free drug, and consequently to increase its anti-cancer activity, a different liposomal formulation was developed [58]. The authors demonstrated that the liposomes protected ATRA and increased its anti-proliferative properties due to the improvement of its cellular uptake, becoming a useful formulation for the treatment of anaplastic thyroid carcinoma [58]. Moreover, Kawakami and colleagues demonstrated that ATRA incorporated into cationic liposomes was efficiently internalized into ATRA-resistant human lung cancer cells in vitro [59]. Specifically, the interaction between the positive charges of the liposomes and the negative charges of the tumor cell membranes allowed the specific internalization of ATRA, thus overcoming tumor cell resistance and producing pro-apoptotic and cytotoxic effects [59].

As already mentioned in the introduction, fenretinide (4-HPR) is a synthetic retinoic acid derivative, endowed with anti-tumor properties and characterized by favorable pharmacological profile, with lower systemic toxicity and better tissue distribution compared to its natural analogue [24]. Nevertheless, the main limitation for the clinical application of 4-HPR derives from its poor bioavailability [27]. Indeed, plasma levels of 4-HPR in patients receiving the maximum tolerated dose (MTD) of the drug (200 mg) were less than 1 µm, not sufficient to produce the desired anti-tumor effects [27]. To overcome such limitation and, consequently, to improve fenretinide performance, efforts to develop proper DDSs have been made in our laboratory. The results achieved, with particular attention on the use of liposomal fenretinide for neuroectodermal-derived tumors, are summarized below.

#### 2.4.1. Neuroectodermal-Derived Tumors

During embryonic development, the three germ layers, endoderm, mesoderm, and ectoderm [63,64], give rise to all the tissues of the adult [65]. The endoderm creates the respiratory system, the digestive system, and some inner organs, such as the thyroid, the thymus, and the liver [66]. The mesoderm produces the musculoskeletal system, the cardiovascular system and the connective tissues [65]. The ectoderm gives rise to the surface ectoderm, the neural tube, and the neural crest [67]. The surface ectoderm generates the epidermis, the cutaneous annexes, the surface epithelium of the mouth and nose, the anterior pituitary gland, the tooth enamel, and the olfactory/optical/optic placodes [67]. The neural tube and the neural crest constitute the neuroectoderm, the first step in the development of the nervous system [68]. The neural tube produces the brain (rhombencephalon, mesencephalon, and prosencephalon), the spinal cord and the motor neurons, the retina, and the posterior pituitary [68]. The neural crest generates the pigment cells in the skin, the ganglia of the autonomic nervous system, the dorsal root ganglia, the facial cartilage, the pulmonary aortic septum of the developing heart and lungs, the ciliary body of the eye, the parafollicular cells (thyroid C cells) and the adrenal medulla [69,70]. In this contest, mutations in mature cells of neuroectoderm may reactivate the embryonic developmental pathways and initiate oncogenesis [71]. Among neuroectodermal-derived tumors, medullary thyroid carcinoma in the thyroid C cells, pheochromocytoma in the chromaffin cells of the adrenal medulla, ganglioneuroma in peripheral nervous system ganglia, malignant peripheral nerve sheath tumor (MPNST) in Schwann cells, melanoma in melanocytes and NB in sympathoadrenal precursors, are among the most frequently reported [71]. In particular, malignant melanoma, the most lethal form of skin cancer [72], affects adult population, with a mean age at diagnosis of 65 years, representing a clinical challenge. Indeed, although progresses have been obtained by immune check point blockade [73], the prognosis for metastatic and/or refractory disease is poor [74], and new therapeutic interventions appear necessary. NB is the most common extra-cranial solid tumor of pediatric age, and it accounts for approximately 15% of all oncology-related pediatric deaths [75,76]. Significant progress has been made in the cure of NB. However, despite the application of aggressive treatment strategies also including differentiation therapies based on the use of 13-cis retinoic acid, the clinical outcome for high-risk NB patients remains poor [77]. As for melanoma, new therapeutic approaches and more effective drugs are needed.

#### 2.4.2. Development, Characterization, and Functionality of Fenretinide-Loaded Liposomes

Sterically stabilized, namely Stealth Liposomes (SL) encapsulating 4-HPR (SL-HPR) were developed and characterized [61,62]. SL are characterized by long circulation into the bloodstream, thanks to the use of the phospholipid 1,2-Distearoyl-sn-glycero-3-phosphorylethanolamine (DSPE) functionalized with polyethylene glycol (PEG), which confers to the liposomes stealth features. Following this method of composition, the recognition of liposomes by the cells of the reticulum endothelial system is reduced, in turn conferring them an increased blood circulation capability and an enhanced possibility to target the tumor [78,79]. SL-HPR showed a chemo-preventive action in the early stages of rat hepatocarcinogenesis, with anti-proliferative effects and apoptosis induction of initiated cells [60]. Moreover, as reported into details below, the same formulation, additionally decorated with an antibody recognizing the disialoganglioside GD2, exerted potent anti-tumor effects against the GD2-expressing neuroectodermal-derived tumors, melanoma and NB [61,62,80].

Attracted from the good properties demonstrated by 4-HPR and on the basis of its in vitro anti-tumor effect obtained when administered as free agent against NB and melanoma cells [10,11], at the end of 90’s, we focused on the development of SL-HPR. The aim was to enhance its bioavailability in vivo and, consequently, the anti-tumor potential. The idea to entrap 4-HPR into liposomes arose also from the fact that the in vitro anti-tumor effects mentioned above were obtained using a dosage 2–10 times higher than that reachable in vivo [81,82].

#### 2.4.3. Fenretinide-loaded Liposomes for Tumor Targeting

SL-HPR were synthesized following the thin-film hydration method, which consists of making a thin lipid film by organic solvent removal, addition of the dispersion buffer and extrusion through 0.2–0.08 μm-pore size polycarbonate membranes, obtaining homogeneous small liposomes [83,84]. In order to increase tumor targeting, SL-HPR were then decorated with the anti-GD2 antibody (a-GD2 moAb), specifically recognizing the disialoganglioside GD2 highly expressed on the cell surface of melanoma and NB cells, and whose expression on healthy tissues is very limited and restricted to cerebellum and peripheral nerves [85,86]. This new formulation was consequently named GD2-Stealth ImmunoLiposomes (GD2-SIL-HPR) (Figure 2), and its functionality on neuroectodermal-derived tumors was firstly assessed on GD2-positive melanoma cells. By competition experiments, cellular association resulted specific and 10 to 15-fold higher compared to that obtained by the untargeted formulation [61]. From a therapeutic point of view, cell proliferation assays revealed that GD2-SIL-HPR were more effective in inhibiting melanoma cell proliferation, compared to both free 4-HPR and SL-HPR. As a note of worth, the anti-tumor efficacy was strictly dependent on the GD2 expression. Indeed, experiments performed on melanoma cells with different degree of GD2 expression showed different response rates to GD2-SIL-HPR treatment, with high response achieved in cell lines expressing high levels of GD2 [61].

The anti-tumor functionality of GD2-SIL-HPR was then evaluated, and extended, against GD2-expressing NB models. Again, in tumor cell binding and uptake studies, the formulation resulted highly selective for GD2-expressing NB cells [62]. Compared to non-targeted liposomes, cellular association of GD2 targeted SIL-HPR resulted 10- to 20-fold higher. As an important note about the carrier integrity, this formulation was able to maintain its ability to bind to GD2-positive cells for at least one week, when stored at 4 °C [61,62].

The in vitro anti-tumor efficacy of GD2-SIL-HPR was studied through cell proliferation experiments, which demonstrated that this formulation significantly inhibited NB cell proliferation compared to HPR, either free or encapsulated in untargeted liposomes. Also in this case, the degree of cell proliferation inhibition correlated with the extent of GD2 expression. NB cell lines expressing higher amount of GD2 better responded to the treatment with GD2-SIL-HPR [62]. A further confirmation of the specificity of the anti-tumor efficacy derived from the treatment of cell lines not expressing GD2. Cell proliferation of these cells was not affected by GD2-tageted-SIL-HPR [62].

Furthermore, to give a translational relevance to the findings obtained in vitro, the GD2-targeted liposomal formulation of HPR was then tested in a metastatic mouse model of human NB, where NBs cells are injected intravenously in the mice tail [87]. This model is clinically relevant, because it mimics both the dissemination of the disease at distant sites, characteristic of patients affected by stage M high-risk NB, and a state of minimal residual disease [87,88]. Treatment with GD2-SIL-HPR determined a complete inhibition of tumor micrometastases development in liver, kidney and ovaries, compared to mice receiving placebo (vehicle buffer only) or free HPR [62]. Survival studies confirmed and supported the above findings. While free HPR and SL-HPR treatments showed a partial anti-tumor response, GD2-SIL-HPR led to long-term survivors, paving the way for future clinical application [62]. 

#### 2.4.4. Fenretinide-Loaded Liposomes for Tumor Vasculature Targeting

A further study conducted in our laboratory, was aimed at developing and testing a liposomal formulation of HPR, specifically targeted to the aminopeptidase N (CD13) receptor expressed by the neo-angiogenic vessels of solid tumors, including NB [33,89,90]. The idea to target tumor vessels within the mass of a solid NB tumor, arises from two main observations; first, high vascular index in NB is associated to poor prognosis [91,92]; second, therapeutic strategy based on the use of liposomal vectors decorated with tumor vasculature-homing and -penetrating peptides might help anti-cancer drugs to overcome many of the physiological barriers present in the abnormal tumor vasculature and on its interstitial matrix, likely increasing their therapeutic effects [34,93].

The new HPR-entrapping, sterically stabilized liposomal formulation, called hereafter NanoLiposomes-HPR (NL-HPR) was developed according to the reverse phase evaporation method [94] and its chemical and structural properties compared to those belonging to the liposomes synthesized and developed for tumor targeting, and following the thin-film hydration method. The reverse phase evaporation method was chosen with the aim to optimize the previously synthesized formulation and because the method used to prepare liposomes can also significantly impact on the physicochemical properties of the payload itself. Size of liposomes was here determined by the dynamic light scattering technology. Compared to SL-HPR, the average size of NL-HPR was smaller (179 ± 4 nm vs 142 ± 3 nm, respectively) and the HPR trapping efficiency was slightly increased (62 ± 7% vs 69 ± 5%, respectively) (Figure 2**)**. The two formulations were also compared in terms of integrity of the vehicle. Importantly, leakage experiments demonstrated that NL-HPR were able to retain the entrapped drug for longer time with respect to SL-HPR. Specifically, while about 50% of HPR was released from SL-HPR in a time of 5 days, the same percentage of release from NL-HPR was reached in 14 days (Figure 2). In summary, the reverse phase evaporation method ameliorated all the liposomal features, rendering the new formulation potentially more performing. 

NL-HPR were then decorated with an NGR-motif containing peptide that specifically targets CD13-positive neo-angiogenic tumor blood vessels [95] (Figure 2). Of note, coupling on the external surface of the targeting moiety did not alter the physicochemical properties of the resulting NGR-NL-HPR formulation [80].

NL-HPR and NGR-NL-HPR were then tested in the clinically relevant orthotopic mouse model of human NB, where a primary tumor mass and metastases at distant sites grow after tumor cells injection in the adrenal gland of mice [96,97]. When compared to control mice and those treated with free HPR, both NL-HPR and NGR-NL-HPR determined a significant increase in life span. However, NGR-NL-HPR led to a significant increase in life span if compared to the untargeted formulation, highlighting the impact of active targeting on the achieved anti-tumor effects. As a consequence, NGR-NL-HPR was the only therapy able to lead a 20% of long-term survivors [80]. These results were explained by the histological analyses performed on tumor samples derived from control and treated animals. As assessed by the staining with the proliferation marker Ki-67, NGR-NL-HPR determined a significant reduction of NB tumor growth compared to both free HPR and NL-HPR. In line with the known mechanism of action of HPR as apoptosis inducer [10,98], TUNEL assays performed on tissue sections from treated mice revealed that the percentage of apoptotic cells was drastically increased when HPR was delivered through the NGR-decorated liposomal formulation. Finally, immunofluorescence staining performed by using anti-CD31, -αSMA, -VEGF, -MMP-2 and -MMP-9 antibodies demonstrated a significant reduction of tumor angiogenesis and capability of tumor invasion, when HPR was administered through NL-HPR and, at a greater extent, through NGR-NL-HPR [80].

## 3. Discussion

Retinoids are a class of natural and synthetic compounds derived from vitamin A, which have raised interest within the scientific community thanks, among others, to their beneficial effects in cancer [7,8,9]. More specifically, the synthetic retinoic acid derivative fenretinide (4-HPR) is endowed with anti-tumor properties and characterized by a favorable pharmacological profile, with lower systemic toxicity and better tissue distribution compared to its natural analog [24]. However, despite substantial in vitro cytotoxicity, response rates in early clinical trials with 4-HPR have been less than anticipated, likely as the consequence of its low bioavailability. To improve bioavailability of 4-HPR, oral powder (LYM-X-SORB^®^, LXS) and intravenous lipid emulsion (ILE) formulations have been tested in early-phase clinical trials [27]. On the other hand, improving pharmacological properties and circulating half-life of retinoids while minimizing their off-target uptake, several delivery systems have been recently developed.

Here, the physicochemical features and the anti-tumor potential of 2 different, sterically stabilized, fenretinide-loaded liposomal formulations developed in our laboratory are summarized. Specifically, these two formulations were different from each other both for the method of preparation (thin-lipid film and reverse phase methods) and for the ligand associated at their external surface to obtain liposomes with the ability to selectively target either the tumor or the tumor vasculature of the neuroectodermal-derived melanoma and neuroblastoma (NB) tumors. Both formulations were found to have optimal physicochemical characteristics, allowing for improving the half-life of the encapsulated 4-HPR drug. Of note, the tumor vasculature-targeted formulation developed following the reverse phase method resulted slightly better than the formulation developed according to the thin-lipid film method in terms of size, polydispersity, drug loading efficiency and drug retention. Indeed, unlike the thin-lipid film method, in which the drug is added after the formation of the liposomes [61,62], the reverse phase method basically involves the initial mixing of the lipids with the fenretinide, the subsequent sonication, and the final evaporation. In this way, more drug was trapped both within the lipid core and at the level of the lipid bilayer itself (see Figure 2), making the vasculature-targeted nanoliposomes more drug-laden, and with a greater retention capacity. However, both 4-HPR-loaded formulations have been shown to exert a potent anti-tumor effect, both in vitro and in vivo. This therapeutic achievement, significantly higher than that obtained by the use of 4-HPR administered in free form or encapsulated in liposomes without targeting, might allow hypothesizing their clinical use in the future.

The use of liposomes as drug delivery has the main purpose of increasing the pharmacological half-life and, at the same time, trying to enhance the tumor-targeting, compared to that obtained by using the same drug but administered in free form. Regarding the delivery of retinoids by liposomes, this will not represent the panacea for all ills, nor will it totally abolish possible side effects. On the other hand, in the last 20 years, we were able to demonstrate that targeted liposomes can ameliorate the drug’s half-life and increase its therapeutic potential [34,87,96,99,100,101,102,103,104]. Moreover, targeted liposomal formulations are well tolerated, and the reduced side effects, compared to free drugs, were demonstrated [105].

## Figures and Tables

**Figure 1 pharmaceuticals-14-00854-f001:**
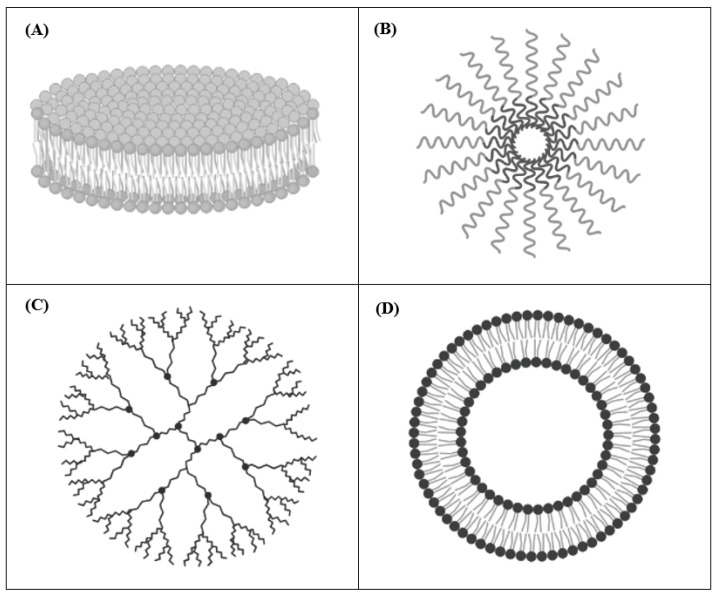
Generic structure of nanodisks (**A**), polymeric micelles (**B**), dendrimers (**C**), and liposomes (**D**).

**Figure 2 pharmaceuticals-14-00854-f002:**
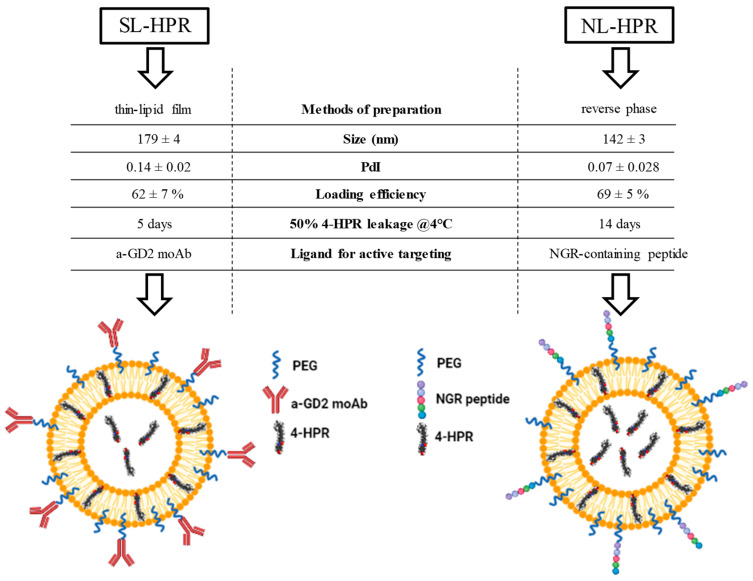
Comparison of tumor- (**left**) and tumor vasculature- (**right**) targeted liposomal formulations of fenretinide. PEG: polyethylene glycol; a-GD2 moAb: anti-disialoganglioside GD2 monoclonal antibody; 4-HPR: N-(4-hydroxyphenyl)retinamide (fenretinide); NGR peptide: NGR-containing peptide; SL: Stealth Liposomes; NL: NanoLiposomes.

**Table 1 pharmaceuticals-14-00854-t001:** Retinoids classification: retinoids are classified into four generations based on their chemical modification [20,21,22].

	First Generation	Second Generation	Third Generation	Fourth Generation
**Features**	▪ Non aromatics ▪ Naturally occurring ▪ Modifications in the polar end group	▪ Mono-aromatics ▪ Modifications in the cyclic ring ▪ More lipophilics and bioavailable	▪ Polyaromatics ▪ Cyclized polyene side chain	▪ Pyranones
**Examples**	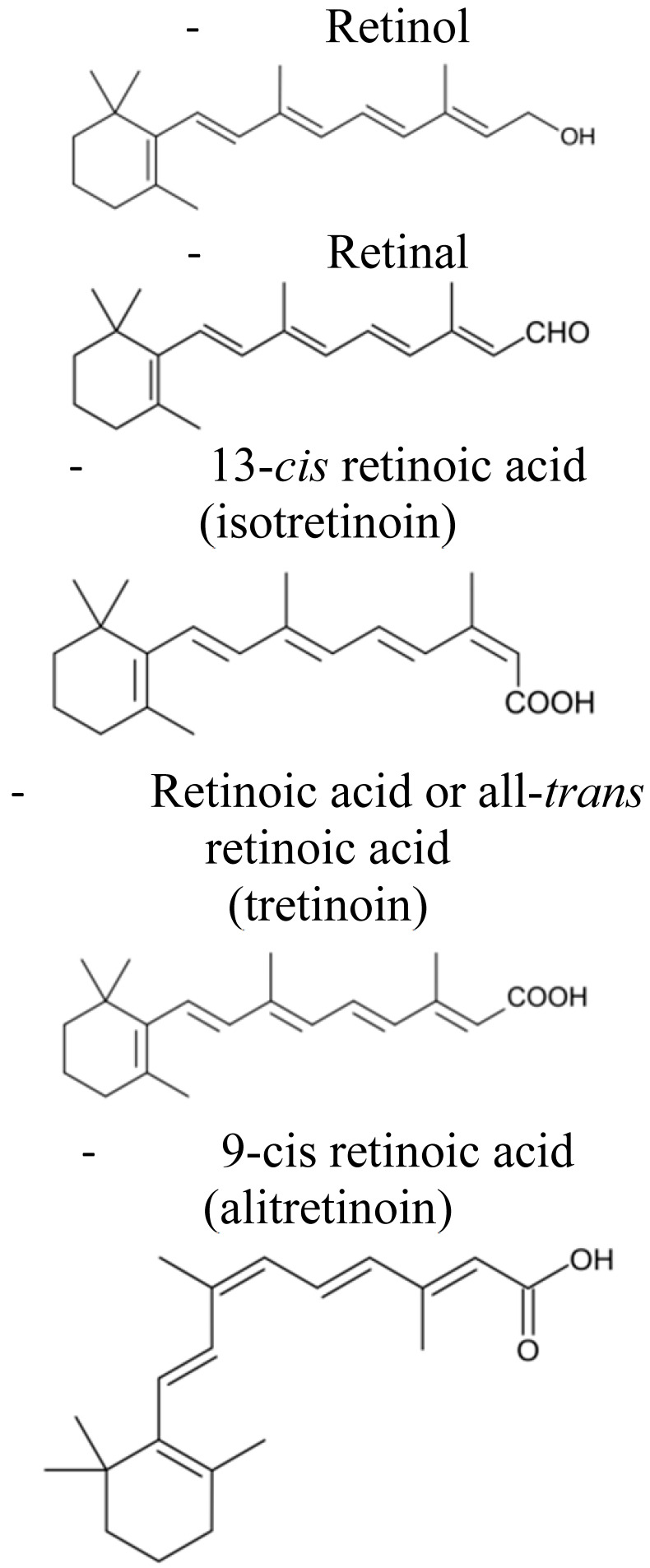	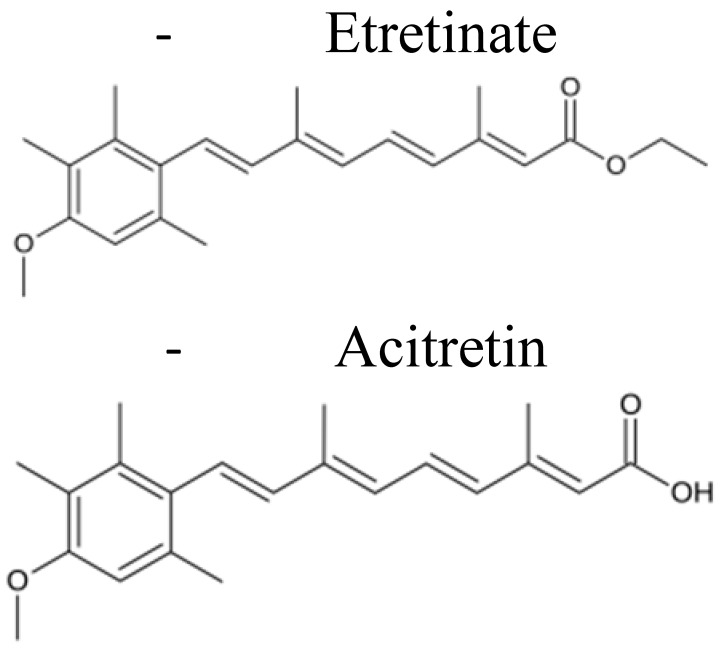	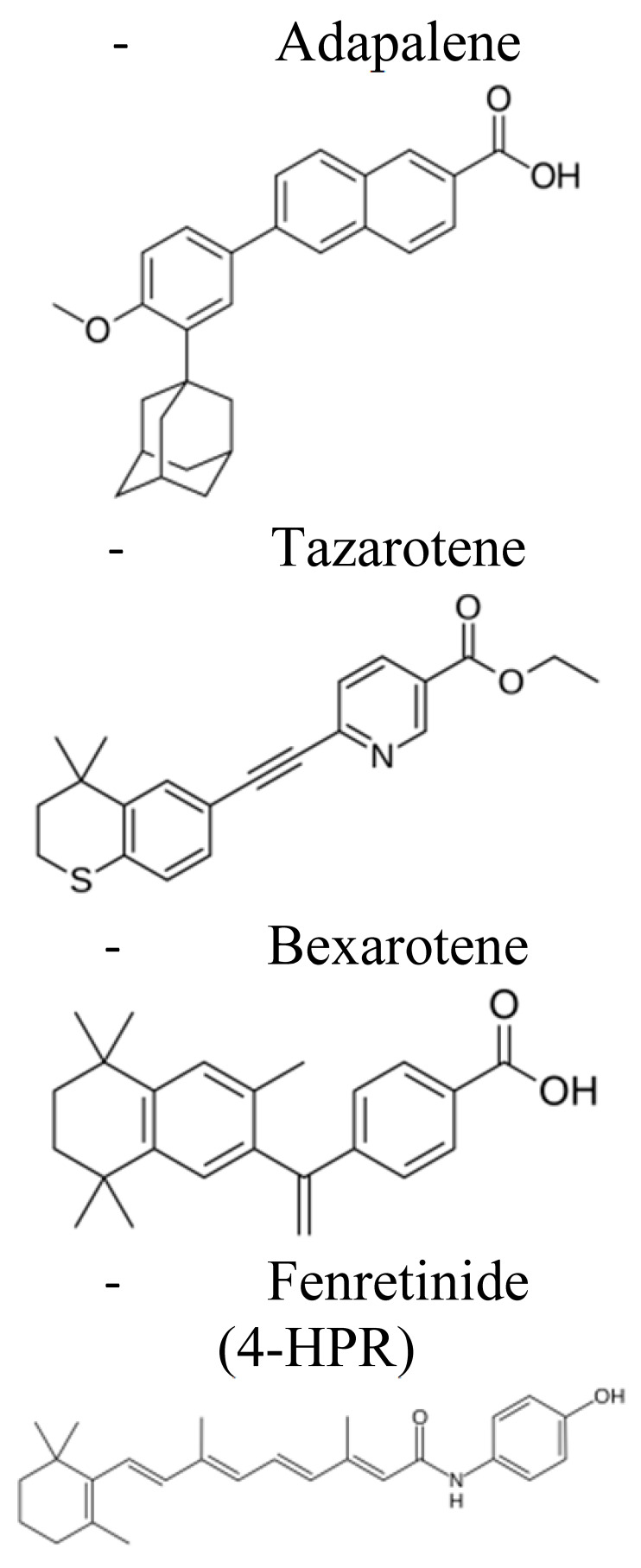	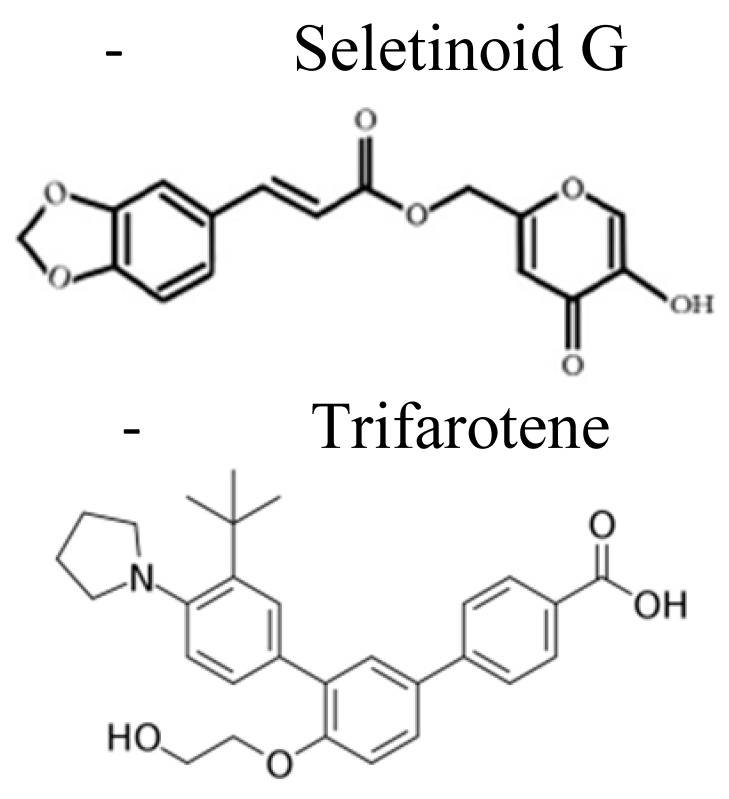

## Data Availability

Data sharing not applicable.

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
