# Peer review of "Retinoids Delivery Systems in Cancer: Liposomal Fenretinide for Neuroectodermal-Derived Tumors"

_pharmaceuticals, 2021, doi:10.3390/ph14090854_

Round 1

Reviewer 1 Report

This review focuses on retinoids delivery systems in cancer. In particular, therapeutic use of fenretinide-loaded liposomes against neuroectodermal-derived tumors such as melanoma and neuroblastoma are discussed. Retinoids have therapeutic potential for treatment of several types of cancer and skin disorders. However, their physicochemical properties such as poor solubility and chemical instability in water, sensitivity to light, heat and oxygen, and low bioavailability limit for clinical translation. This review shows a strategy to overcome the drawback of retinoids for therapeutic applications by using nano-carriers such as liposomes. This topic is suitable for the journal. My review comments are as follows,

  1. Page 7, line 10 and Page 8, Figure 2 caption; “polyethilenglicol” should be polyethylene glycol.
  2. Page 7, line 40; extrusion through polycarbonate membranes. Which pore size of the polycarbonate membranes was used?
  3. Page 8, middle paragraph; “As an important note about the carrier integrity, this formulation was able to maintain its ability to bind to GD2-positive cells over time, when stored at 4°C [61,62].” This sentence is not clear to know how long the binding ability is maintained.
  4. Page 9, middle paragraph; “NL-HPR was smaller (179±4 vs 142±3, respectively)” These values should have a unit (nm).
  5. Why liposomes prepared by the reverse phase evaporation method are able to retain the entrapped 4-HPR for longer time than liposomes prepared by thin-lipid film method? Please discuss this point in more detail in manuscript as this point is important in aspects of pharmaceutics.
  6. Nano-carriers would drastically change the bioavailability and biodistribution of the retinoids. Even if the nano-carriers have targeting properties, the main organs for uptake of the typical nano-carriers are liver, spleen, and bone marrow, those have mononuclear phagocyte system. By modifying the bioavailability and biodistribution of the retinoids using nano-carriers, the side-effects such as liver toxicity might be enhanced. Authors should discuss the possible side-effects of nano-carrier system.
  7. In this review, 2 different fenretinide-loaded liposomal formulations are focused. These two formulations were prepared by different methods. In conclusion, authors describe that “a sterically stabilized, tumor vasculature-targeted, 4-HPR-loaded nanoformulation showed excellent physicochemical features, allowing to hypothesize its clinical use in the future.” This conclusion sounds like a conclusion only from the physicochemical aspect of liposomes prepared by the reverse phase evaporation method. Because two formulations are designed for different targeting mechanisms, tumor-targeting and tumor vasculature-targeting, authors should lead the conclusion including in vitro and in vivo therapeutic performances of tumor-targeting and tumor vasculature-targeting for retinoids delivery.

Author Response

Genoa, August 23th, 2021

Guess Editor

Pharmaceuticals Editorial Office

Dear Dr. Carol Liu,

Thank you for your email of August 19th, 2021, with the decision regarding our manuscript entitled “Retinoids delivery systems in cancer: liposomal fenretinide for
neuroectodermal-derived tumors”  (Manuscript ID: pharmaceuticals-1350647).

As you suggested, please find attached a revised version of the manuscript: as requested, we have addressed all reviewers’ comments. Changes are highlighted in yellow in the main manuscript file. Moreover, following the Academic Editor request, we expanded the body text of the manuscript, both through the responses to the reviewers, and by adding a paragraph on tissues and tumors of neuroectodermal origin.

The reviewers' comments have been addressed as follows:

Reviewer #1:

  1. We thank the Reviewer for the helpful comments. Below, point-to-point answers to the raised doubts and requests are shown.

    Page 7, line 10 and Page 8, Figure 2 caption; “polyethilenglicol” should be polyethylene glycol.

AAs. We thank the Reviewer for the correct comment. Misspelled words have been correctly modified.

    Page 7, line 40; extrusion through polycarbonate membranes. Which pore size of the polycarbonate membranes was used?

AAs. We thank the Reviewer for the question. The pore size of the polycarbonate membranes used have been added within the text.

    Page 8, middle paragraph; “As an important note about the carrier integrity, this formulation was able to maintain its ability to bind to GD2-positive cells over time, when stored at 4°C [61,62].” This sentence is not clear to know how long the binding ability is maintained.

AAs. We thank the Reviewer for the interesting comment. The requested information have been added within the text.

    Page 9, middle paragraph; “NL-HPR was smaller (179±4 vs 142±3, respectively)” These values should have a unit (nm).

AAs. We thank the Reviewer for the correct comment. The unit of measurement have been added.

    Why liposomes prepared by the reverse phase evaporation method are able to retain the entrapped 4-HPR for longer time than liposomes prepared by thin-lipid film method? Please discuss this point in more detail in manuscript as this point is important in aspects of pharmaceutics.

AAs. We thank the Reviewer for the interesting question. Fenretinide-loaded nanoliposomes were prepared through the reverse phase method. Unlike the thin-lipid film method, in which the drug is added after the formation of the liposomes, the reverse phase method basically involves the initial mixing of the lipids with the fenretinide, the subsequent sonication and the final evaporation. In this way, more drug is trapped both within the lipid core and at the level of the lipid bilayer itself, making the vasculature-targeted nanoliposomes more drug-laden, and with a greater retention capacity. This comment was added in the “Discussion” Section.

    Nano-carriers would drastically change the bioavailability and biodistribution of the retinoids. Even if the nano-carriers have targeting properties, the main organs for uptake of the typical nano-carriers are liver, spleen, and bone marrow, those have mononuclear phagocyte system. By modifying the bioavailability and biodistribution of the retinoids using nano-carriers, the side-effects such as liver toxicity might be enhanced. Authors should discuss the possible side-effects of nano-carrier system.

AAs. We are in agreement with the reviewer’s comment. The use of liposomes as drug delivery has the main purpose of increasing the pharmacological half-life and, at the same time, trying to enhance the tumor targeting, compared to that obtained by using the same drug but administered in free form. Undoubtedly, the use of liposomes for delivering retinoids will not be the panacea for all ills, nor it will totally abolish possible side effects. On the other hand, in the last 20 years we were able to demonstrate that targeted liposomes can ameliorate the drug's half-life and increase its therapeutic potential (Pastorino F et al, Cancer Res 63: 86-92, 2003; Pastorino F et al, Cancer Res 63: 7400-7400, 2003; Pastorino F et al, Clinical Cancer Res 2003; Pastorino F et al, Cancer Res 2006; Loi M et al, J Control Rel 2010; Di Paolo D et al, Molecular Therapy 2010; Loi M et al, J Control Rel 2013; Pastorino F et al, Small 2019; Di Paolo D et al, Small 2020). Moreover, we have already demonstrated that, targeted liposomal formulations are well tolerated and reduce side effects compared to free drugs. For instance, In Cossu I et al paper (Biomaterials 2015), the anti-tumor effects of targeted liposomes encapsulating DXR were validated in an orthotopic model of NB using a dedicated micro-PET system. The analysis indicated that targeted liposomes were able to delay tumor growth compared to control or free-DXR-treated mice. The observed decrease in tumor size was paralleled by a reduction in tumor glucose consumption. Interestingly, the combined analysis of cancer and whole body glucose metabolism confirmed the selectivity of DXR-loaded targeted liposomes action. Indeed, targeted liposomal DXR did not modify blood clearance of FDG, which was, instead, significantly increased by free-DXR, indicating that free-DXR modified the systemic metabolic pattern due either directly or triggering stress-mediated hormonal response. Accordingly, from the pharmacological and the toxicological points of view, these findings support the concept of a higher tolerability of DXR-loaded liposomal formulation with respect to conventional (free) drug administration.

Part of this comment was added into the “Conclusion” section.

In this review, 2 different fenretinide-loaded liposomal formulations are focused. These two formulations were prepared by different methods. In conclusion, authors describe that “a sterically stabilized, tumor vasculature-targeted, 4-HPR-loaded nanoformulation showed excellent physicochemical features, allowing to hypothesize its clinical use in the future.” This conclusion sounds like a conclusion only from the physicochemical aspect of liposomes prepared by the reverse phase evaporation method. Because two formulations are designed for different targeting mechanisms, tumor-targeting and tumor vasculature-targeting, authors should lead the conclusion including in vitro and in vivo therapeutic performances of tumor-targeting and tumor vasculature-targeting for retinoids delivery.

AAs. We thank the Reviewer for the interesting comment. The last part of the “Conclusions” Section, now called “Discussion” has been modified as follow:

Here, the physicochemical features and the anti-tumor potential of 2 different, sterically stabilized, fenretinide-loaded liposomal formulations developed in our laboratory are summarized. Specifically, these 2 formulations were different from each other both for the method of preparation (thin-lipid film and reverse phase methods) and for the ligand associated at their external surface to obtain liposomes with the ability to selectively target either the tumor or the tumor vasculature of the neuroectodermal-derived melanoma and neuroblastoma (NB) tumors. Both formulations were found to have optimal physicochemical characteristics, allowing for the improvement of the half-life of the encapsulated 4-HPR drug. Of note, the tumor vasculature-targeted formulation developed following the reverse phase method resulted slightly better than the formulation developed according to the thin-lipid film method, in terms of size, polydispersity, drug loading efficiency and drug retention.  Indeed, unlike the thin-lipid film method, in which the drug is added after the formation of the liposomes, the reverse phase method basically involves the initial mixing of the lipids with the fenretinide, the subsequent sonication and the final evaporation. In this way, more drug is trapped both within the lipid core and at the level of the lipid bilayer itself, making the vasculature-targeted nanoliposomes more drug-laden, and with a greater retention capacity. However, both 4-HPR-loaded formulations have been shown to exert a potent anti-tumor effect, both in vitro and in vivo. This therapeutic achievement, significantly higher than that obtained by the use of 4-HPR administered in free form or encapsulated in liposomes without targeting, might allow to hypothesize their clinical use in the future.

Reviewer 2 Report

The manuscript entitled “Retinoids delivery systems in cancer: liposomal fenretinide for neuroectodermal-derived tumors” is well-written. Nevertheless, some minor corcerns should be adressed before being published in Pharmaceuticals.

RARs and RXRs should be defined.

The quality of the figures inserted into Table 1 must be improved.

“The most relevant DDSs used in pre-clinical studies of cancer therapy are nanodisks (NDs) and micro/nano-particles, such as polymeric micelles, dendrimers and liposomes (Figure 1)” The reviewer suggests to the authors not to mix the term nanoparticle with that of other systems because it may confuse the reader.

“Moreover, liposomes are the only nanoparticles used in clinical trials for the delivery of retinoids in solid cancer” The reviewer suggests: moreover, liposomes are the only nanosystems…..

The authors should include units for size in Figure 2.

“The HPR trapping efficiency was slightly increased” The results obtained are very similar. Therefore, the reviewer disagrees with this statement.

Author Response

Genoa, August 23th, 2021

Guess Editor

Pharmaceuticals Editorial Office

Dear Dr. Carol Liu,

Thank you for your email of August 19th, 2021, with the decision regarding our manuscript entitled “Retinoids delivery systems in cancer: liposomal fenretinide for
neuroectodermal-derived tumors”  (Manuscript ID: pharmaceuticals-1350647).

As you suggested, please find attached a revised version of the manuscript: as requested, we have addressed all reviewers’ comments. Changes are highlighted in yellow in the main manuscript file. Moreover, following the Academic Editor request, we expanded the body text of the manuscript, both through the responses to the reviewers, and by adding a paragraph on tissues and tumors of neuroectodermal origin.

The reviewers' comments have been addressed as follows:

Reviewer #2:

  1. We thank the Reviewer for the helpful comments. Below, point-to-point answers to the raised doubts and requests are shown.

RARs and RXRs should be defined.

  1. In agreement with the reviewer, RARs and RXRs have been defined.

The quality of the figures inserted into Table 1 must be improved.

  1. In agreement with the reviewer, The quality of the figures inserted into Table 1 have been improved.

“The most relevant DDSs used in pre-clinical studies of cancer therapy are nanodisks (NDs) and micro/nano-particles, such as polymeric micelles, dendrimers and liposomes (Figure 1)” The reviewer suggests to the authors not to mix the term nanoparticle with that of other systems because it may confuse the reader.

AAs. We thank the Reviewer for the correct comment. The proper sentence has been modified as follow:

The most relevant DDSs used in pre-clinical studies of cancer therapy are nanodisks (NDs) polymeric micelles, dendrimers and liposomes (Figure 1).

“Moreover, liposomes are the only nanoparticles used in clinical trials for the delivery of retinoids in solid cancer” The reviewer suggests: moreover, liposomes are the only nanosystems…..

AAs. We thank the Reviewer for the correct comment. The sentence was modified as suggested.

The authors should include units for size in Figure 2.

AAs. We thank the Reviewer for the correct comment. The unit of measurement have been added.

“The HPR trapping efficiency was slightly increased” The results obtained are very similar. Therefore, the reviewer disagrees with this statement.

AAs. We can understand the Reviewer comment. We, however, believe that the adverb “slightly” is appropriate in this case, when the HPR trapping efficiency between the 2 formulations was not so significant, but still different.